# An Integrated Assessment of the Horticulture Sector in Northern Australia to Inform Future Development

**Kamaljit K. Sangha** [1,*], **Ronju Ahammad** [1], **Muhammed Sohail Mazahar** [2], **Matt Hall** [2], **Greg Owens** [3], **Leanne Kruss** [4], **Gordon Verrall** [5], **Jo Moro** [6] and **Geoff Dickinson** [7]

1   Research Institute for the Environment and Livelihoods, Charles Darwin University, Casuarina, NT 0810, Australia
2   Department of Industry, Tourism and Trade of the Northern Territory Government, Darwin, NT 0800, Australia
3   NT Farmers Association, Coolalinga, NT 0810, Australia
4   Queensland Agriculture Workforce Network, Brisbane, QLD 4003, Australia
5   Department of Primary Industries and Regional Development, Perth, WA 6000, Australia
6   FarNorth Queensland (FNQ) Growers, Cairns, QLD 4870, Australia
7   Horticulture & Forestry Science, Department of Agriculture and Fisheries, Brisbane City, QLD 4000, Australia
*   Correspondence: kamaljit.sangha@cdu.edu.au

**Abstract:** The horticulture sector in northern Australia, covering north of Western Australia (WA), Northern Territory (NT), and north Queensland (QLD), contributes $1.6 billion/year to the Australian economy by supplying diverse food commodities to meet domestic and international demand. To date, the Australian Government has funded several studies on developing the north's agriculture sector, but these primarily focused on land and water resources and omitted an integrated, on-ground feasibility analysis for including farmers'/growers' perspectives. This study is the first of its kind in the north for offering a detailed integrated assessment, highlighting farmers' perspectives on the current state of the north's horticulture sector, and related challenges and opportunities. For this, we applied a bottom-up approach to inform future agriculture development in the region, involving a detailed literature review and conducting several focus group workshops with growers and experts from government organisations, growers' associations, and regional development agencies. We identified several key local issues pertaining to crop production, availability of, and secure access to, land and water resources, and workforce and marketing arrangements (i.e., transport or processing facilities, export opportunities, biosecurity protocols, and the role of the retailers/supermarkets) that affect the economic viability and future expansion of the sector across the region. For example, the availability of the workforce (skilled and general) has been a challenge across the north since the start of the COVID-19 pandemic in 2020. Similarly, long-distance travel for farm produce due to a lack of processing and export facilities in the north restricts future farm developments. Any major investment should be aligned with growers' interests. This research highlights the importance of understanding and incorporating local growers' and researchers' perspectives, applying a bottom-up approach, when planning policies and programs for future development, especially for the horticulture sector in northern Australia and other similar regions across the globe where policy makers' perspectives may differ from farmers.

**Keywords:** agriculture; horticulture; economy; food; markets; northern Australia; research and development; supply chain; workforce

## 1. Introduction

Northern Australia, covering an area of 3 million km$^2$ across the Northern Territory (NT), northern parts of Western Australia (WA), and north Queensland (QLD), supports approximately 5% of the total Australian population (~1 million people) [1,2]. Beef production, mining, conservation, and agriculture (horticulture, broadacre cropping, and

plantation forestry) are the main land uses in the region (Figure 1), with pastoral use being the dominant sector across all jurisdictions [3,4]. Over half of the landmass in this region possesses a legal recognition of Indigenous interests (freehold and native title), and the rest supports freehold or lease tenure arrangements for diverse land uses [5].

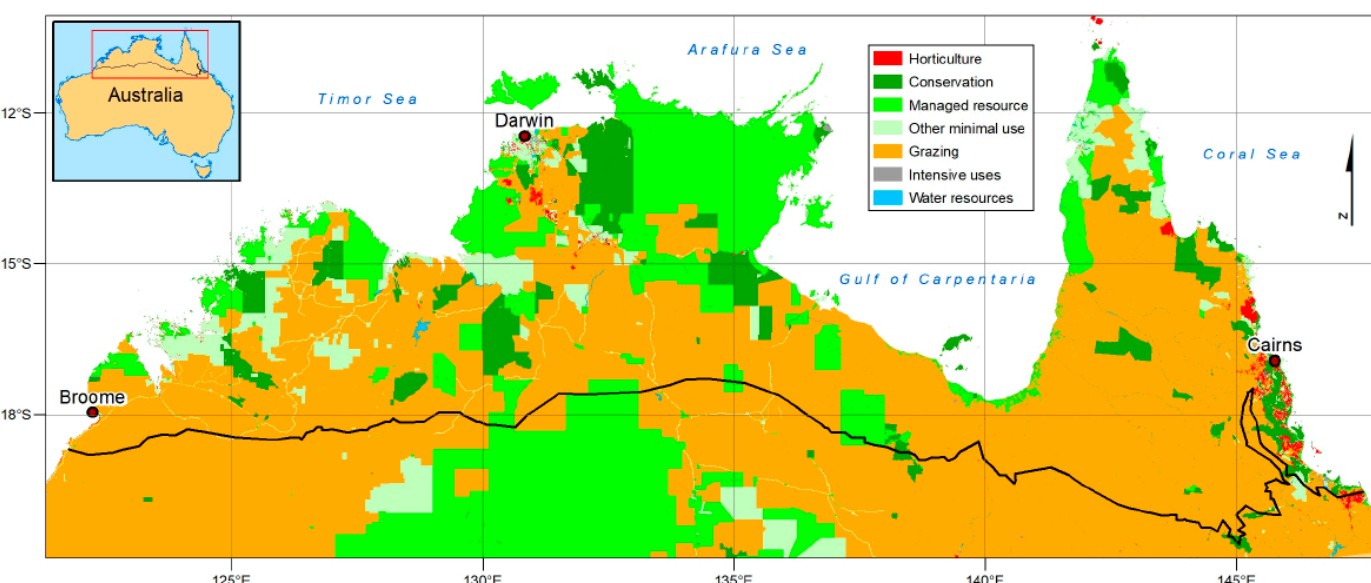

**Figure 1.** Main land uses across northern Australia (red areas indicate agriculture/horticulture). (Courtesy Andrew Edwards).

Northern Australia has often been tagged as under-developed for its remote and rural economies [3]. To develop the north, the Australian Government has led a few initiatives in the past, mainly to support mainstream economies such as mining and agriculture [6]. Under the current 'Developing the North' agenda [1], the government plans to invest substantially to improve infrastructure and facilities, and is committed to supporting and developing the agriculture sector across the north. Some significant developments have already commenced such as establishing the Northern Australia Infrastructure Facility (NAIF), Cooperative Research Centre for Developing the Northern Australia (CRCNA) to support research investment, upgrading sea freight facilities in Darwin, and aquaculture expansion in the NT (project Sea Dragon—a Prawn farm) and north Qld (a Prawn farm in Proserpine). The availability of 17 million hectares of arable soil [7] could further support about 1.84 million hectares of irrigated agriculture across the north [8,9]. However, basic information on what kind of agriculture development farmers want, and the current status of horticulture in the north, including challenges and realistic opportunities from farmers' perspectives, is lacking for all the northern jurisdictions—a gap that our research addressed in this paper.

Northern Australia offers some unique and competitive advantages over other regions for 'out of season' production of exotic fruits and vegetables (e.g., early-season mango and melon production), a suited tropical climate for Asian fruits and vegetables, and a vast area of land for farm expansion [3]. The high seasonality of rainfall with a significant variation from <250 mm to >4000 mm per annum also offers a wide range of farming options across the region [10], which includes a variety of exotic tropical and sub-temperate fruits and vegetables, and nursery production to meet the domestic as well as international demands of food and lifestyle products. The volume of production and economic value of some highly demanded fruits in Australia such as avocados, bananas, lychees, mangoes, and melons are confined largely to the north given the topical climatic conditions. These products often dominate the Australian markets and also receive demand from the international market for their high quality [11].

Over recent years, the horticulture sector has maintained a steady production and economic growth in the north [2,3]. The region, being proximal to Asia where the socio-economic capacity of the middle class has significantly improved in recent years, offers a great opportunity to export high-quality commodities, such as mango, lychees, and avocados [12]. For instance, KPMG (2020) estimated the export potential at $120 million by 2030, depending on the availability of adequate logistic facilities for air and sea freight in north Qld. Expanding irrigated horticulture in the north to meet the growing demand for premium and value-added commodities in the Asia–Pacific region, while benefiting local communities, is a priority under the 'Developing the North' agenda [1].

The development of the horticulture sector across northern Australia can also contribute towards achieving United Nations Sustainable Development Goals 2 (zero hunger) and 3 (good health and well-being) [13], for improving health and providing a nutritious diet at affordable prices, not just for people in Australia but also in the entire Asia–Pacific region [14]. However, an integrated understanding of the north's social, economic, environmental, and climatic contexts is essential to realize such outcomes and sustain them.

Several studies [11,12,15,16] have recognised a range of challenges associated with developing the north, but from a broader agricultural and not focused horticultural perspective. A few studies that focus on the horticulture sector describe only a single or highly valued commodity (e.g., mango or avocado) and related supply chains for domestic and international markets [2,11], and omit the details about the sector itself. An integrated understanding of the current situation of the north's horticulture sector, and of related challenges and opportunities, particularly from growers' perspective, to cognise the local situation, and explore new, feasible, opportunities and related markets to develop the sector, is lacking to date. This study aimed to address these fundamental gaps to inform the development of the north's horticulture industry including research and future investment, by appropriately informing decision-making at the State/Territory as well as at the national level. We synthesise and present on-ground information from farmers, researchers, industry experts, and others representing each State and Territory in northern Australia to inform sustainable planning and development for the north's horticulture industry.

This study can inform similar agriculture developments happening in many parts of the developing world where farmers are deemed 'uneducated' and their perspectives are often not valued and ignored. We stress here that farmers, who work with their land and develop production systems, hold immense knowledge and skills and they should be part of the decision-making process when it comes to agriculture-related development. This study offers a pathway to how that is achievable.

## 2. Methodology

### 2.1. Brief Background

This research is part of a CRCNA funded project "A situational analysis of the horticulture sector in northern Australia" [17,18]. The aim of the project is to understand the diversity, complexity, challenges, opportunities, and threats for the horticulture sector in northern Australia, and to inform future policies, research and development, and investment, for developing ecologically sustainable, climate-smart, and economically viable horticulture systems in the north. The geographical region, covering more than one-third area of the Australian continent, as defined by CRCNA (Figure 2), was selected for the study.

### 2.2. Approach for Data Collection and Analysis

This research applied a variety of methods for data collection including literature review, focus group workshops, and online survey questionnaires with farmers/growers and experts from the industry and research bodies related to horticulture.

To understand the status and issues affecting the growth and future development of the horticulture sector in the region, firstly we reviewed literature available from the Australian Bureau of Statistics (ABS), Australian Bureau of Agriculture and Resource Economics and Sciences (ABARES), various government organisations in the NT, Qld

and WA, Farmers Associations, Hort Innovation, peer-reviewed scientific journals, and unpublished reports. The main attributes for this review included identifying the type of major commodities, and their production volume and area, economic value, and any issues related to the north. Data from various sources were reviewed and collated, which laid the foundation for discussion with the study participants on defined themes (e.g., production, supply chain, workforce, etc.—details below) developed by the research team.

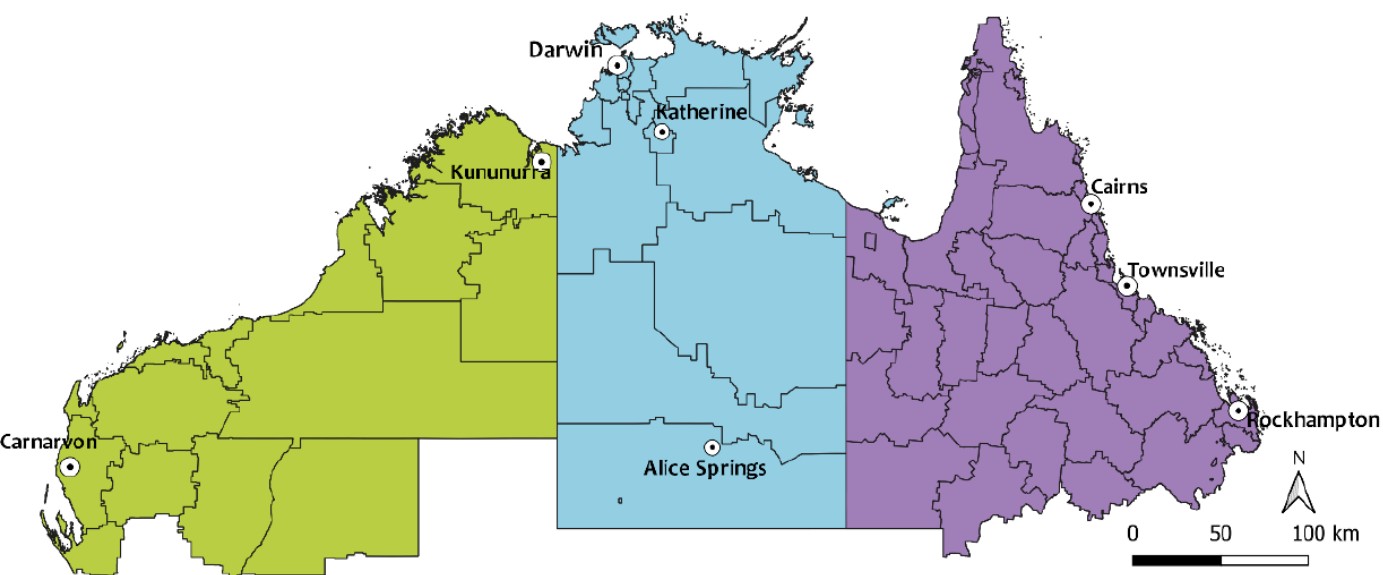

**Figure 2.** Major horticulture centres (indicated by black dots) across northern Australia.

Following the literature review, we conducted focus group workshops with growers from the NT, north Qld, and north WA, and industry experts from the region (details in [18]). The participant growers represented major industries such as mango, banana, melon, and other tropical fruits and vegetables. The expert participants were represented by the government horticulture research, development and extension, and regional development agencies, i.e., the Department of Industry, Tourism and Trade (DITT) of the NT, Department of Primary Industries and Regional Development (DPIRD) of WA, Department for Agriculture and Fisheries (DAF) of Qld, Hort Innovation Australia, and consultants working on the production, marketing, supply chains, and export.

Focus group workshops were of two types: (i) three workshops mainly with growers from each northern state/territory to understand state-level issues; and (ii) a main expert workshop including both growers, and industry and market representatives from across the north to understand and analyse the situation from a broader northern perspective. Each workshop comprised between 7 and 15 participants. Discussions were held on defined themes. The state/territory level workshops in the NT and Qld were held face-to-face, except for WA. The experts' workshop was of mixed mode, with 12 participants face-to-face who were split into two groups of six each, and three online (who could not travel due to COVID-19 restrictions). In all workshops, a structured discussion was followed covering the following defined themes: production and economic efficiency; supply chain; workforce; research and development; and policies/institutional support. For each theme, we applied the "Strengths (S), Weaknesses (W), Opportunities (O), and Threats (T)—SWOT" analysis approach to explore the growers' and experts' perspectives. SWOT analysis is commonly used for analysing and positioning an industry's resources and planning [19]. With this approach, we were able to critically analyse the current state, challenges, and opportunities for the horticulture industry.

For the experts' workshop, 12 participants joined the workshop in person and 3 online from Qld and WA. The research team facilitated online and face-to-face discussions. The workshop material was communicated to each participant in advance. During the workshop,

three online participants formed one group, and after discussion among themselves, they presented and shared their views with the face-to-face group. The online group also later collated their responses using an MS Word document which was sent to the researchers.

For the WA workshop, the research team organized the workshop in collaboration with DPIRD researchers, and growers were invited to the DPIRD office to be together to participate and discuss issues, challenges, and opportunities that they experience. The research team organised prior trial sessions with DPIRD researchers to familiarise them with the workshop topics and format, and a workshop agenda and outline were sent in advance. On the workshop day, the research team facilitated discussion using the online zoom platform.

In addition to focus group discussions on the defined themes, the participants were given a semi-structured questionnaire for each group (both online and face-to-face) to suggest their agreed opinions for each theme on a Likert scale, i.e., very low (1) to very high (5). For instance, to assess the domestic market, the participants were asked to discuss among themselves and collectively respond to the question "how would you rate the demand for northern produce in the domestic market of Australia?". This was then followed by three associated questions on the domestic market considering the "weaknesses", "threats", and "opportunities for improvement". For the supply chain, we asked the participants, for example, "how effective is the existing supply chain across the north?" and a few related questions on transport facilities, costs, market protocols, and consistency in the supply chain. In this way, we were able to deliberately lead discussions on each of the topical issues to derive conclusions.

Overall, we triangulated information from literature review, focus group workshops, and semi-structured group (and online) questionnaires. Focus group workshops delivered mostly qualitative information in the forms of text (i.e., statements, notes) and group surveys presented agreed responses for a specific topic/issue on a Likert scale: very low (1) to very high (5) with a mix of quantitative and qualitative responses. The data were re-checked with the participants from focus group workshops to re-confirm their views. Subsequently, we analysed and collated information in a summary matrix following SWOT.

## 3. Results

We firstly present an overview of the performance of the northern horticulture industry, followed by the current state and challenges as experienced and explained by the participants, and then the sector-related opportunities, as below.

### 3.1. An Overview of the Performance of the Northern Horticulture Industry

Northern Australia produces a diversity of fruits and vegetables which include avocados, bananas, citrus, mangoes, melons, and a variety of vegetables (pumpkins, sweet corns, tomatoes, etc.), mostly produced in north Qld followed by the NT and northern WA. Bananas represent a major horticultural commodity, accounting for the largest share of all horticultural produce in quantity and value, with 336,000 tons of production in north Qld (2019–2020). The second major commodity is water- and musk- melons, with 66,000 tons in the NT; 65,000 tons of production in north Qld; and 21,000 tons in north WA. Mangoes comprise the third-largest horticulture commodity, with 32,000 tons of production in 2019–2020 in the NT, and 22,000 tons in north Qld (Figures 3 and 4). Compared to the NT and north Qld, the quantity of horticulture produce in north WA is relatively small (Figure 3).

In terms of the economic value of produce, during 2019–2020 bananas in north Qld were worth $583 m/yr, followed by mangoes in the NT worth $129 m/yr (fetching a much higher price than mangoes from north Qld ($49 m), Figure 3), and tomatoes in north Qld ($87 m/yr) (Figure 3). Melons were worth $69 m/yr in the NT, $59 m/yr in north Qld, and $20 m/yr in north WA (2019–2020). Avocados in north Qld, a fast-growing economic commodity, were worth $63 m/yr. Among vegetables, sweet corn and beans in north Qld represented the highest value, i.e., $76 m/yr and $66 m/yr, respectively. In the NT, vegetables (mostly Asian) were worth $61 m/yr, and in north Qld other vegetables

(including capsicum, potato, pumpkin, etc.) were worth $124 m/yr (excluding tomatoes and beans).

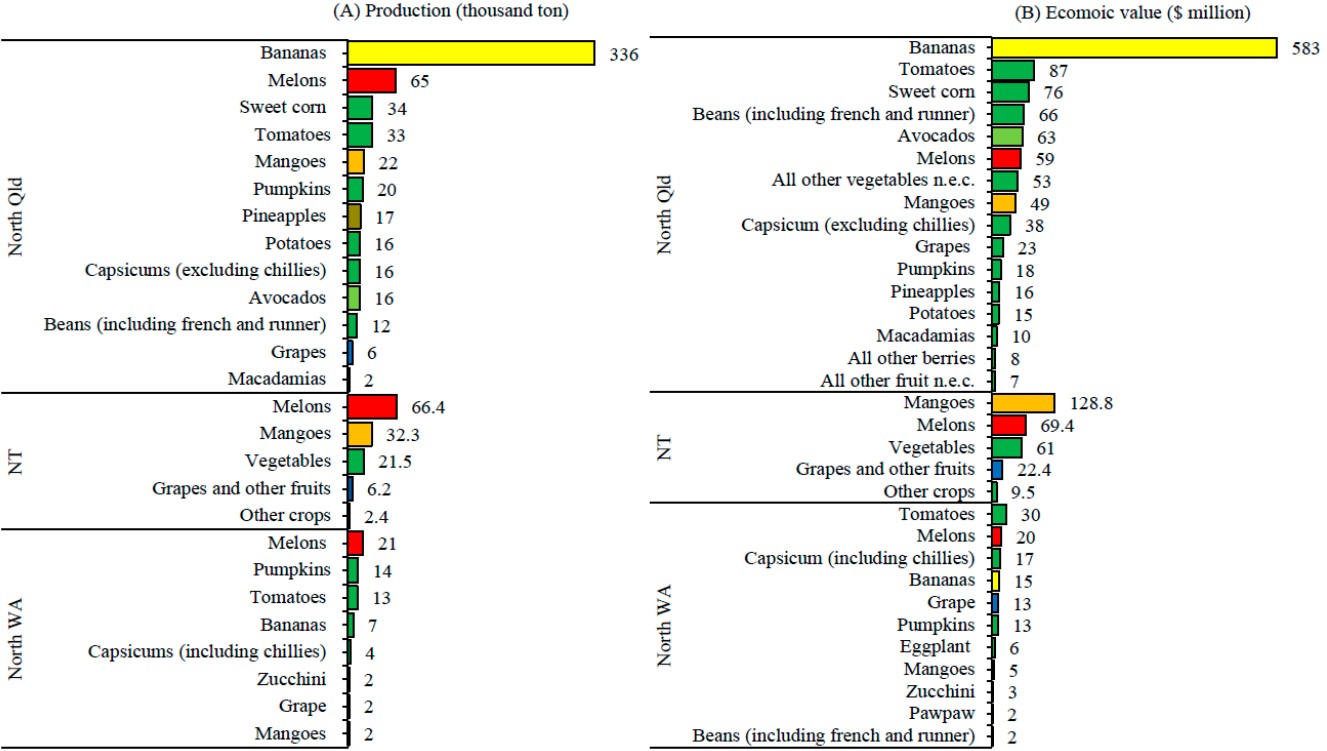

**Figure 3.** Estimate of the production (**A**) and economic values (**B**) of the horticulture commodities produced in northern Australia (north Qld, NT, and north WA) for 2019–2020 [20–23].

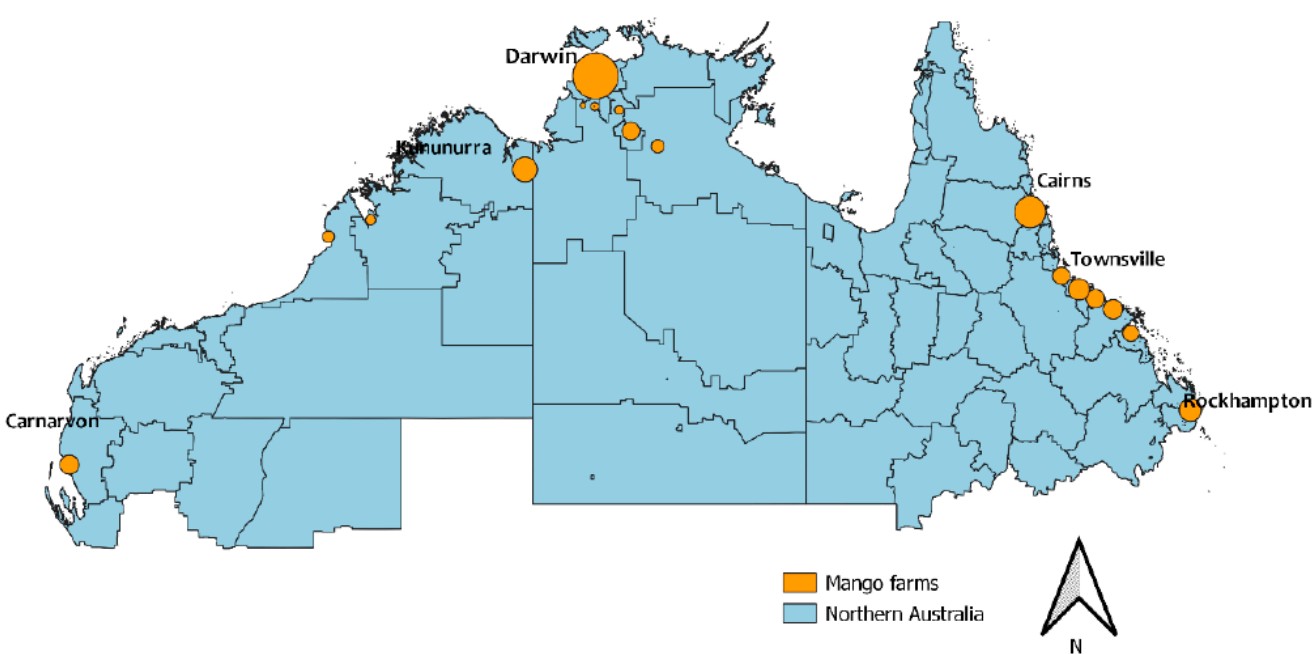

**Figure 4.** Main areas of mango farming in northern Australia (modified from Applied Agricultural Remote Sensing Centre (AARSC) [24].

The net value of major horticultural produce across northern Australia is worth $1.6 b/yr (2019–2020) where north Qld supports $1.2 b/yr, the NT $291 m/yr, and north

WA \$126 m/yr. These estimates exclude several emerging crops such as durian, jackfruit, lychee, and other tropical commodities due to a lack of confidence in data sourced from the ABS and Hort Innovation.

### 3.2. Current State and Challenges for the Horticulture Sector of Northern Australia

Our study offers insights on horticulture production and the availability of land and water resources; important issues and challenges related to the market supply chain; the workforce (general and skilled); and research development and extension across the region, as presented below.

### 3.2.1. Production and the Availability of Land and Water Resources

Respondents suggested that the overall performance of commodities such as mangoes, melons, bananas, and vegetables (well-established industries) is stable across all jurisdictions of northern Australia. At a state level, in general, production levels are relatively high in north Qld, steadily improving in the NT, but relatively low in north WA. At a farm level, respondents identified climatic factors, for instance, an increase in temperature negatively affecting mango flowering and subsequently its production across the north. Since the existing crop varieties are lacking an improved genetic pool over recent years, the respondents highlighted that the threat of disease outbreaks and climatic variability are likely to be more frequent and predominant in the future—exhibiting major threats to the sector. There is a lack of long-term data on how an increase in temperature will impact mango production across the region, which is a dominant, high-value, commodity in the region. Similarly, the banana industry is highly impacted by climate change, particularly the increased frequency of cyclones and flooding in northern Qld. In particular, respondents indicated that the effects of extreme weather events, i.e., excessively hot and windy conditions, on the horticulture systems are already evident in northern parts of WA (Kununurra and Carnarvon), north Qld, and the Top End, NT. In general, across the north, the horticulture systems are quite efficient but increasing input costs mainly for fertilizers, pesticides and transport constantly challenge the sustainability and profitability of the sector (Supplementary Materials: Tables S1–S4).

In terms of market economy, high-demand exotic crops such as avocados, lychees, mangoes, etc. afford reasonable returns. However, a strong influence of retailers and supermarkets in the Australian markets means commodity prices are determined by the retailers rather than growers (Supplementary Materials: Table S1, for further details see [18]). As a result of market mechanisms, the growers receive a much lower price (i.e., 9–10 fold less) than what the consumers pay in the market (Tables S1 and S2). This declining marginal profitability is exerting increasing pressure on many small- and medium-scale farmers who are left with no other options than to sell their farms to corporations. This situation is especially common in north Qld (Table S2). Growers expressed serious concerns about the future of their family farms.

Water availability and accessibility for irrigation were reported as key factors to maintaining existing production as well as determining the expansion of farms across the north. Most discussants agreed that an adequate supply of water is currently available for irrigation for the existing farms across all the northern states and territories; however, variability exists among jurisdictions depending upon the source of water. For example, in the NT, growers largely rely on groundwater for irrigation. Allocation of groundwater is currently restricting the availability of water for further horticulture expansion in the region (Table S1). The participants reported that about 40%–50% of the allocated water remains unused in the Douglas and Daly region in NT (only 25%–50% is effectively used in farming). There is little understanding of the impacts of surface water extraction and of long-term use of ground-water resources for expanding or developing new farms in the NT. However, growers in north Qld and northern WA use relatively less groundwater and more surface water (i.e., river dam). In WA, participants reported that irrigation techniques are still underdeveloped and inadequate to expand horticulture in the Ord region (Table S1).

In north Qld, the Tinaroo dam supplies surface water for irrigation to >700 farms (Table S2; also see [18]). Due to high rainfall in the region, the water supply has remained relatively stable to date. However, there is no alternative source for water security, especially in the event of declining rainfall and increasing climate change, particularly when the horticulture industry (worth > $1 b/yr) depends on the only dam in the area (Table S2).

Land availability is not a major concern, as reported by the participants. However, the accessibility and useability of land for expanding horticulture is a significant challenge across all northern jurisdictions (Tables S1–S4). Each state/territory of northern Australia has a different set of land legislations. Typically, Qld legislation supports relatively easy access to cultivable land, especially with large areas under the crown or pastoral lease and/or freehold land. In the NT, currently small areas of land are available under freehold or crown lease, particularly around Darwin and Katherine. Most of the NT's land is registered as inalienable aboriginal freehold land (~50%, 63 m ha) under Aboriginal Land Rights Act 1976, and the rest as other land parcels (mainly pastoral or crown lease). Similarly, in WA, significant parcels of land are registered under Aboriginal Land Trust, and the rest, predominantly, under the pastoral lease. Across all those leases, Native Title and Indigenous Land Use Agreements (ILUA; mostly prevalent in Qld) are the only common legislations across all northern jurisdictions. The respondents emphasized that the current policy and regulation barriers further impede the acquisition and/or access to land for horticulture development. Underdeveloped pastoral areas are also often difficult to convert to alternative land uses such as horticulture due to the high level of investment required for developing road infrastructure and other facilities on farms. Furthermore, any investment for farm expansion requires a consistent and secure water supply to make land productive. Participants, particularly from north Qld, emphasized that the need is to develop sustainable and economically viable farming systems rather than expanding or clearing more land for farming unless export markets are explored and supported by the governments (Tables S1 and S2). Overall, the process to negotiate land tenure and access to land and water resources is quite complex and expensive in WA and the NT, as reported by the participants.

### 3.2.2. Market Supply Chain

The supply chain comprises all the stages and activities associated with the production from the farm gate to the consumers. The market demand for existing horticultural produce from northern Australia was revealed as 'very high' by the participants; however, the activities involved in getting produce from the farm gate to the consumers involve significant costs for transportation and refrigeration due to greater distances (Supplementary Materials: Tables S1–S4). The current state of the supply chain includes several steps, each with associated costs for growers (e.g., transport facilities and costs, uploading, storage, treatment, unloading, etc.), which affect producers' economic profitability (Table S1). Maintaining consistent temperature as required for different commodities over long distances is critical for delivering a good quality product in the market; however, it is a significant issue for long-distance transport. With the availability of data loggers, there have been significant improvements over the last 4–5 years.

The current market system also demands consistency in supplying a particular commodity in large volumes which is sometimes difficult for growers to commit to, particularly under changing climate conditions (Table S1). Currently, there is no stock take of the quantity and quality of even main commodities in a region. Overall, the participants ranked the state of the supply chain as 'medium to highly effective' in the north. Among various jurisdictions, it is relatively more effective in Qld, followed by the NT, and then northern WA (Supplementary Materials: Table S1).

The state of existing transportation facilities in terms of availability of trucks and transportation costs were ranked as moderate to high in the north. Within the region, transport facilities are relatively efficient in north Qld, followed by the NT and northern WA (Table S1). Due to large distances, the NT and WA growers pay high transportation

costs. One of the reasons identified by the discussants is that there is little competition among the road freight companies in the NT and northern WA (e.g., Kununurra) as growers rely on a few transport companies only. Typically, the volume of produce and time of the year determine the availability of the freight and costs, which works in favour of northern growers due to 'out of season' production compared to their southern partners. In this regard, high-demand commodities (e.g., mango, banana, etc.) have good options for transportation. However, product inconsistency, small volumes, and variable supply of some commodities such as jackfruit, durian, rambutan, etc. result in higher costs for growers in the NT and north WA. Small-scale growers, particularly in Asian vegetables, are significantly affected due to limited and expensive transport facilities as they often aggregate their multiple commodities in the same freight to save costs (Tables S1–S4). In addition, there are no opportunities in the north for value-adding to the local produce.

Lack of awareness among growers about the supply chain is another issue reported by many participants (Table S1). Most growers run family businesses and often lack the ability to understand the market situation or changes in demand and consumer behaviour. Typically, growers are disconnected from consumers, which results in a lack of understanding among the growers to adapt their products to consumer demand. For example, in the case of emerging crops (e.g., carambola, jackfruit, dragon fruit, etc.), very limited information is available from the retail shops on the health benefits or how to process a product to aid customers in decision-making for new commodities. Currently, Asian vegetables and fruits are produced at a small scale, without any proper inventory, just to meet the domestic market demand. A detailed assessment can help Asian vegetable and fruit growers to understand the market situation, address the market needs, and adapt their produce as per consumer needs (Table S1).

Another concern raised by the participants in the supply chain is a lack of uniform market protocols across different Australian states and territories (Table S1). This affects the economic viability of the sector. Each state has its own biosecurity policies, regulations, and compliance procedures, and growers must fully comply with those protocols for selling their produce in the target market of a state/territory. Inconsistent protocols incur significant costs to growers as they adapt to different states/territories' procedures. At present, growers of high-value products such as mangoes, due to their volume and consistent supply, can minimise the costs of compliance, whereas the small-scale producers experience huge costs for complying with the protocols, and sometimes lack the ability to understand these requirements. These compliance procedures sometimes also lack updates, for example, irradiation treatment is currently available to enhance product shelf-life without any damage, and is particularly useful for tropical fruits and vegetables; however, retailers in southern markets are not willing to adopt it because of consumer concerns (Table S1). We acknowledge that there are few producers across the north who take the advantage of the state/territory-specific protocols (e.g., specific treatment to sell lemons/limes in the WA market) as they adapt their product to meet the needs of a particular market, and prefer that targeted market approach.

Overall, key challenges in the supply chain include greater distances and transport costs for freight, lack of cooling before transportation and monitoring of input temperature during transport, and a lack of awareness among growers and consumers about the whereabouts and the quality of produce (branding, etc.), lack of stock take of main commodities at a regional level, and variability in the state/territory compliance protocols for accessing and selling produce in various southern state markets—these collectively affect the economic efficiency of the northern farming systems.

### 3.2.3. Workforce (Harvesting and Skilled)

All participants highlighted that the availability of seasonal workforce to harvest horticulture produce has become a major concern, especially under COVID-19 conditions, which was not of concern prior to the COVID-19 outbreak (Supplementary Materials: Tables S1–S4). Since most horticulture produce is seasonal by nature, casual workforce,

in high numbers, is required but only for short periods. Due to the ongoing pandemic, the labour shortage is exacerbated by visa changes, leading to limited migrant labour. Participants also identified other issues such as remoteness of the farms, inadequate accommodation, and language barriers. By and large, farm management within the horticulture sector is also not up to date to consider changing demand and strategies for workforce retention (Table S1). Human Resources management systems exist only on very few large farms whereas most medium- and small-scale farms maintain these skills on a seasonal basis, without a reasonable investment in their workforce. Many growers who rely on middlemen/workforce contractors to access the workforce also experienced extra costs. There is a lack of workforce strategies across the sector. Investing in developing workforce strategies could assist state/territory and farmers while demonstrating how they meet workforce needs. NT Farmers Association recently developed an NT Plant Industries Workforce Development Plan that covers some of the aspects mentioned earlier, including listing the required workforce in technologies such as robotics, database management of the farming systems, precision agriculture, etc.

Over the past years, growers have demonstrated efficiency in managing their farms, knowledge, and skill transfer among their own family members across the north. However, growers expressed a sober concern around the future of the family farms with a declining interest among the young generation, and the need for agronomic, managerial, and technological skills required on the farms (Table S2). The lack of interest among the young Australians to work on the farm was raised as a serious matter by all the participants, which was partly attributed to the poor public image of the sector. Currently, at the college and university levels, only a few options exist for young Australians to pursue career pathways in the agriculture/horticulture sector. Most small and family growers cannot afford resources to support such pathways for their children (except for large farms or corporations); thus, often youth is attracted to take on more rewarding employment options (e.g., mining) than agriculture. The majority of the participants, especially family growers, showed their willingness to support training opportunities for the new workforce to build their knowledge and skills on the farms (Tables S1–S4).

### 3.2.4. Research, Development, and Extension

The participants suggested that, in general, existing research support for the horticulture sector is adequate to improve farming systems, domestic and international market access, and to understand the supply chain (Table S1). A key issue, as mentioned by the group, is a weak linkage between the Research and Development Corporations (RDCs) and field interventions, particularly for enabling growers to target commercial production outcomes. A bulk amount of research has been completed in the past years; however, it has not always been extended and adopted at the farm level. To date, most of the research has been conducted in north Qld, but not in other areas such as WA, and the central and southern parts of the NT. There is a lack of understanding of the commercial reality of emerging crops in terms of economic viability, demand, supply chain, and sustainable market access. Besides, adequate research has only been undertaken on big commodities such as mangoes, not on small-scale emerging crops (e.g., Asian vegetables and fruits) (Table S1). Additionally, climate change-related research has only covered a few main crops, and it is yet to be translated for the producers to understand. Due to the intensity and changing frequency of severe weather events, the respondents asserted that the growing negative impacts of climate change on other crops should be studied more (Table S1).

The respondents reported that Hort Innovation is one of the key horticulture RDCs that has not fully focused on grower-driven strategies for development or innovations in the north. For instance, market access for northern producers is a serious concern that has not been fully addressed to date, partly because of a lack of collective efforts by the RDCs, Hort Innovation, and state governments. Although Hort Innovation represents over 40 industries, some emerging crops afford limited levy generation with little support mechanisms for research, development and extension such as the Asian vegetables. Often

the national priorities of the industry peak bodies do not always match with local and regional needs in the north, which leads to mis-informing development policies at a local scale. Another major issue is that most research and development-related investment occurs in disrupted, often short, phases due to short-term government grants and a lack of long-term commitment (Table S1).

### 3.3. Opportunities for Improving the Horticulture Sector of Northern Australia

The participants identified several key areas to improve the northern horticulture sector. These include efficiency and transparency in market price, access, and supply chain; availability of workforce; availability and accessibility (and security) of land and water; and applied research focusing on development and extension to address growers' needs (Table S1; further details in [18]). On a farm scale, increasing the size of the business and improving regularity in commodity supply can benefit small growers (Table S1). Since there are different types and sizes of commodities produced across the north, a flexible transport system with multi-chambered refrigeration arrangements to accommodate a variety of commodities is essential. Improving rail connectivity, the number of trucks, and service (chambered refrigeration with controlled temperature) could improve the market supply of tropical fruits and vegetables from the north, as suggested by the participants.

Generally, horticulture growers are efficient, yet their capacity to understand the complex supply chain processes and related updates need to be enhanced for operating a successful enterprise. A holistic understanding of both production and marketing processes may enable them to oversee production and focus more on managing the supply of their commodities. Largely, the growers have to accept the conditions defined by the retailers/wholesalers, and deliver supplies without much control (Table S2). Building growers' capacity to understand and interpret market conditions is critical to strengthening their capacity to advance their businesses. It will also enable them to identify specialized markets (e.g., organic produce) or to grow crops that do not need storage or ripening. Some growers with emerging crops (i.e., jackfruit, durian, carambola) also need a better way of promoting their produce to attract and raise awareness among customers about the nutrient qualities of tropical fruits and vegetables (Table S1).

To address the workforce issues in the north, diversification of large family farms to engage workforce throughout the year could offer long-term solutions, as suggested by the participants (Table S1). This will require new capital investment for upscaling the enterprise. Additionally, building relationships among different businesses may provide options for sharing the workforce. At the farm level, a significant investment is required to create livable conditions for the workforce through provisioning accommodation, transport, etc., which are generally beyond the capacity of small growers. In this regard, participants suggested that the industry or farmers' associations can play a critical role to find collective solutions, and governments can promote seasonal work programme (e.g., Pacific Labour Scheme). The growing demand for a skilled workforce in the sector offers another opportunity to train young Australians in VET and agriculture-related courses in universities to meet the industry needs and enhance the application of technology in farming systems.

Developing strategies and policy frameworks on sustainable use of land and water resources, especially where more than one party has an interest (such as Traditional Owners with Native Title rights and leaseholders), offers diverse opportunities to consider innovative projects such as Indigenous bush food. The current state of knowledge also requires water modelling to determine surface water capacity, update the regulatory system, and explore trading mechanisms to enhance the efficient use of water resources. A critical opportunity, as identified by the participants, also exists to facilitate genuine engagement processes with Indigenous peoples to collectively plan for developing the horticulture systems in the north.

Assisting and informing farmers to understand new information such as the impacts of climate change, the availability of new agriculture technologies that may help growers to adapt and be efficient, is stressed as a key priority by the participants. At the market

level, finding cost-effective and simple protocols across the states/territories to avoid any unnecessary burden on growers as well as retailers is another challenge. In this regard, collaboration among the diverse group of growers should be strengthened, which is currently restricted within a commodity group, rather than across the commodities (Table S1).

With the ongoing COVID-19 pandemic since the start of 2020 and its unforeseen disruptions to workforce, crop harvest, and the market supply chain, the respondents stressed the need to develop a disaster resilience and strategic capacity management system to build resilience of the north's horticulture industry (Table S1). They emphasised developing a future disaster management plan while considering the cyclic nature of disastrous events that the north's horticulture sector has experienced over the past years. There are opportunities to learn from and develop adaptive plans across the production and supply chain processes. For example, the Panama disease outbreak in bananas in north Qld has resulted not only in economic loss but also psychological stress among the growers. About 30% of losses in banana production were attributed to labour shortage, as the respondents mentioned, and the rest to many other social factors. The group suggested that it is time to assess the domestic capacity in terms of existing risk management strategies, what works well, and the gaps across the sector so that future risk management strategies focus on building resilience across the sector. Developing a cohesive and integrated resilience system, embracing diverse fruits and vegetables across the north, offers a novel opportunity that can help the farming systems to plan and adapt for any disasters, in advance. A major challenge for this research is to convey growers' perspectives to policy decision-makers so the appropriate strategies and solutions could be developed and implemented for supporting development that is sustainable over the long-term.

## 4. Discussion

Horticulture in northern Australia offers a diverse range of exotic fruits and vegetables with a relatively stable supply for the Australian domestic market, the total value of the produce being $1.6 billion per annum. This research, for the first time, applied a bottom-up approach to explore the main strengths, weaknesses, opportunities, and challenges of the sector by involving a wide range of growers, industry experts, and researchers from each northern jurisdiction. In particular, growers' associations were collaborators in this research and played a key role to highlight growers' perspectives on the current issues, challenges, and opportunities that exist to expand the sector. To date, several scientific reports conducted by the Commonwealth Scientific and Industrial Research Organisation (CSIRO; [8,10]) and others have mainly projected scientific perspectives of expanding horticulture/agriculture in the north. Conversely, this study outlined growers' and local experts' perspectives from each northern jurisdiction, providing a realistic picture that can inform future investment and development of the sector. We argue that when it comes to development that has a direct impact on farming systems, farmers should be made part of the decision-making process to convey their perspectives. Similarly, across the globe often farmers' views are omitted or neglected—this approach should be changed to embrace the rich and on-ground knowledge that many farmers bring to the table which can help policy decision makers to develop realistic projects, in line with local aspirations.

A key common issue identified by the participants across all northern jurisdictions is a lack of data specific to the local/regional areas, particularly for north Qld—a major producer in the region. This local scale-specific data are vital to inform future development and investment in the horticulture sector. Data were also lacking on emerging Asian fruits and vegetables. For example, several types of Asian vegetables and exotic tropical fruits, largely produced in the NT, are not recorded in any government publications [25]. Likewise, Kununurra and Carnarvon in WA jointly produce a majority of bananas, mangoes, melons, eggplants, beans, and zucchini and satisfy the domestic demand in WA, but these are not mentioned in the national reports [20,21]. A thorough understanding of what kind of commodities, and where they are produced, across the north, is much required to inform future research and investment policies for developing the north's horticulture sector.

Another common issue is related to the market pricing where retailers and supermarkets (Coles and Woolworths) exhibit their total control on price and the type of produce that a farmer can sell in the Australian market. Lack of transparency in the current market system, with typically 9–10 fold less price than what produce is sold for in the supermarkets, raises serious concerns about farm economic profitability for growers across the north.

A range of key issues, challenges, and suggestions for improvements, as identified by this research, are discussed below.

The availability of arable lands and their proximity to water sources is fundamental for making any decision to expand the horticulture sector in northern Australia, and several CSIRO reports [8–10] have attempted to address this aspect. However, there is no integrated assessment in terms of water and land resources, land tenure, market opportunity, and the economic feasibility of developing new farming systems. A key challenge is that most of the agricultural land in the north is only moderately suitable for production and most of the area is categorised as marginal [10]. Small areas of suitable or fertile land exist but only in patches [8], raising concerns about the long-term sustainability of any agriculture-related enterprises. Long-term sustainability, particularly for water and land usage in the farming systems was considered of concern by the workshop experts. In terms of water resources, allocation of water across the north is a critical issue, and it may hinder future investments until there is clarity and security of water usage [26,27]. If a water resource is to be shared, given Indigenous people have significant legal rights to land and related resources across the north [4], understanding and developing genuine benefit-sharing mechanisms in consultation with the Indigenous communities is a first step [28]. The need is to understand the long-term availability, security, and impacts of water usage on natural systems in the north [26], which can inform and help design water allocation principles and policies, and related benefit-sharing mechanisms. In north Qld, growers particularly emphasised ensuring the security of water supply for existing farms rather than developing new dams for expanding horticulture (Table S1, [18])—contrary to the Australian Government's 'Developing the North' agenda (2015). Especially, any further expansion of water resources should meet the needs of existing growers, their economic ability to bear the cost, and their willingness to be part of the new irrigation infrastructure.

Land tenure arrangements, currently different across all northern jurisdictions, offer a major challenge that requires consultations with Indigenous peoples who have various legal rights under freehold, exclusive and non-exclusive native titles. Dale and Taylor (2014) suggested that solving land tenure issues is essential. However, not much progress has been made in that direction to date. Ignoring or underestimating Indigenous rights has led to the halting of or threatening some planned agriculture-related developments in the north. For example, development at Singleton Station is challenged by the Traditional Owners of the area for excessive use of water resources [29]. There is a need for caution, and for the developers to understand the nature of water and land resources, and indigenous interests so that sustainable practices are followed to address public and indigenous concerns. Developing specific resource use principles and related policies for different jurisdictions to improve access to land and water for any land development-related projects in the north is much required at present. From our research, growers in north Qld prioritised developing a secure market for their existing commodities than any further land developments, as the latter most likely may benefit the corporates and big producers only while marginalising small-medium scale growers. This contrasting view of growers to the government suggests developing a realistic land development plan in order to create a win–win situation for both corporates and small growers, and for not flooding the existing domestic market of the same commodities. The current approach for development, without exploring export markets, is reducing profit margins and may further marginalise small-scale growers (>300 in north Qld alone) (personal communication with growers in the region). Similarly in the NT, regardless of potential irrigation through surface water capture, access to land may not be easy due to the complicated land tenure process and economic costs involved for site development.

The lack of suitable infrastructure to expand the horticulture industry in the north was highlighted as a key issue by the participants. Long-distance road transport from the north to south and associated costs reduce growers' economic returns, particularly for low-value crops and small-scale growers (i.e., vegetables, jackfruit, durian etc.) [2]. Moreover, there are significant greenhouse gas emissions and related costs, with produce first travelling to southern markets and then some travelling back to the north [30] which typically remain unaccounted for. Strategies are required to promote and develop local markets in the north and export the rest. Significant investment from the private industry and/or government in transport, especially in aviation, ports, rail, digital connectivity, power, and storage (particularly in refrigerated cabins), is necessary to progress the sector in all northern jurisdictions [31]. Proximity to Asia and tropical weather to produce 'out of season' commodities can help realize the huge export potential that has been highlighted by many [1,11,12,32]; however, this requires appropriate export facilities and state-level negotiations to understand the market protocols of target countries [3]. Expansion of export facilities in the north is highlighted by all the participants while recognising that a coordinated approach between businesses and the Australian Federal and State/Territory Governments is vital. This is of high priority for north Qld growers. They suggested that the Cairns airport and seaport can be operationalised with updated facilities as part of the long-term export infrastructure. Likewise, Darwin port could be developed for export purposes for the NT and northern WA growers.

Inconsistent supply chain policies and procedures across various states and territories offer another challenge for northern growers who often lack the time or ability to understand the rapidly changing market. Typically, growers have limited information and opportunities to collaborate and learn about their supply chain [11,33]. To make the northern businesses efficient for saving costs and time, simple and easy biosecurity measures across the states are essential to effectively deliver farm commodities to the wholesalers and retailers [34]. Learning and training in biosecurity protocols will help build growers' confidence in the supply chain, along with implementing simple procedures for diverse tropical products [35]. Likewise, building growers' capacity in the supply chain processes will help them better understand the logistics (i.e., freight and related costs), and changes in market demand and consumer behaviours.

The workforce, especially under COVID-19, has emerged as a serious issue across all the northern jurisdictions. Historically, the low level of human capital has been a key impediment to agricultural developments than climatic or agronomic constraints across the north [3]. The harvesting labour shortage exacerbated by the ongoing COVID-19 outbreak calls for governments to support reliable seasonal work visa programmes, as well as agriculture-focused educational opportunities for the new entrants in the sector [36,37]. A short-term migrant workforce may not offer an optimal solution in the horticulture industry over the long term given several cases of compliance failure at workplaces and the limited time frame for the workers' visa [38,39]. However, farmers' capacity to scale up enterprise (i.e., increase diversity of produce and volume) and apply integrated farm management (i.e., human, technical and managerial) can also contribute to retaining farm labour over a longer period. Adapting to the changing conditions e.g., nighttime harvesting as practised by one mango farmer in the NT, could also help attract domestic labour into farming. Since labour shortage during the harvesting period presents an economic risk (i.e., loss of harvest), growers have expressed keen interest in adopting technological solutions such as robots to pick mangoes, that can save costs and ensure profitable and viable enterprises in the long run [40].

The north offers a huge potential for broadacre crops, pasture and forage, various horticulture commodities, and emerging crops, which, when cultivated in the right combination, could make businesses economically viable in the region [2]. However, there is a lack of integrated environmental, social, and economic feasibility assessment to determine the scope of horticulture developments that can make the industry productive, profitable, and sustainable. All the participants suggested, in unison, that on-ground research, in

consultation with growers, is required to understand key challenges in the farming system including addressing climatic variability, constraints to accessing land and water resources, supply chain and market procedures, and enhancing growers' ability to adopt new technologies. A recent study in the NT [41], demonstrating the impact of climate change on mango flowering and production, has been considered quite useful by the growers. Similar studies are required for other commodities for growers to make future investment decisions. In addition, long-term research projects that can demonstrate the impacts and outcomes over a longer time period could help boost trust for investing in developing the horticulture industry in the north.

This research intends to inform governments across the north by highlighting on-ground growers' perspectives and concerns. It suggests the need to understand 'development' beyond the current paradigm of clearing land and building dams for agriculture expansion. Development in true sense is about enabling people to lead their lives (following [42]), thus supporting many small and medium scale growers who are contributing significantly to sustain the horticulture industry and regional towns and communities across the north is of utmost importance. Moreover, supporting regional communities delivers benefits to many other local people (rather than a few corporates), and hence the local economies. The economic consequences of marginalising small- and medium-scale growers will be serious over the long-term, if not corrected.

Overall, there are several serious challenges for developing the horticulture sector in the north, but they also offer innovative opportunities to explore new commodities, new crop combinations, and export markets. The need is to understand and incorporate growers' and experts' perspectives into decision-making, in north Australia and elsewhere across the globe, and to plan and develop sustainable, climate-smart, resilient farming systems applying integrated, long-term, socio-economic, and environmental perspectives.

## 5. Conclusions

Northern Australia is proposed as the agriculture frontier [1] due to the availability of land, seasonal availability of water, and recent growth in the horticulture industry. This research identified key challenges (i.e., workforce, infrastructure, markets, economic viability, and use of water and land resources) and related opportunities (to grow 'out of season', a diverse range of exotic, tropical crops and vegetables). Significant prospects exist to expand the sector, mainly if the international market for export economies is targeted and developed as the domestic market is already saturated. Each of the identified challenges demands due consideration, involving diverse stakeholders who have an interest in the use and management of land and water resources (e.g., growers, land managers, indigenous land owners and managers, public, government, etc.), which can ultimately help promote sustainable, efficient, and equitable development in the north.

**Supplementary Materials:** The following are available online at https://www.mdpi.com/article/10.3390/su141811647/s1, Table S1. Experts workshop. Table S2. NQld workshop. Table S3. NT workshop. Table S4. WA Kununurra Workshop.

**Author Contributions:** Conceptualization by K.K.S.; methodology by R.A.; workshops by all authors; validation of information by all authors; formal data collection and analysis mainly by R.A., K.K.S. and M.S.M.; writing—original draft preparation by K.K.S. and R.A.; writing—review and editing—all authors; project administration, K.K.S. and funding acquisition, K.K.S. All authors have read and agreed to the published version of the manuscript.

**Funding:** This research was funded by the Cooperative Research Centre for Developing Northern Australia under a project (A.1.2021063).

**Informed Consent Statement:** Informed consent was obtained from all subjects involved in the study.

**Acknowledgments:** We acknowledge the funds provided for this research by the Cooperative Research Centre for Developing Northern Australia (CRC NA). We sincerely thank many growers and experts from across the north who participated in our workshops and contributed immensely to

state/territory level as well regional level discussions. We thank Allan Dale, the Chief Scientist for CRC NA, for his feedback. We also acknowledge the assistance and support rendered by Queensland Department of Agriculture and Fisheries; the Department of Primary Industries and Regional Development, WA; and the Department of Industry, Tourism, and Trade, NT.

**Conflicts of Interest:** The authors declare no conflict of interest.

**Ethics Approval:** The ethics approval for this study was obtained from the CDU Human Ethics Committee (reference no. H22015).

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
