# Peer review of "An Integrated Assessment of the Horticulture Sector in Northern Australia to Inform Future Development"

_sustainability, doi:10.3390/su141811647_

Round 1

Reviewer 1 Report

Interesting paper that addresses topical issues not only local according to mixed approaches not only scientific approach, it analyzes the state of Australian agriculture from different aspects, the issues are expressed quite clearly even if there is a generically lack of data and analyzes that led to these considerations, the text would benefit and the reading would be facilitated. Even the integration of schemes, figures and tables would make the work even less journalistic.

Author Response

Response:

Thanks a lot for your positive feedback. We have included more workshop data as appendices in the revised paper.

Kind regards

Kamaljit

Reviewer 2 Report

I read a very interesting research paper. Congratulations for the hard work! Two major comments:

- Even if some efforts have been made to address sustainable development, it seems that the authors have worked more on economic viability than on sustainability. If this approach is voluntary, they must explain the relevance of such a choice. Is it because the other dimensions of sustainable development are in good shape?

- "This study can inform similar agriculture developments happening in many parts of the developing world where farmers’ perspectives are often not considered and ignored saying them ‘uneducated’." The authors must absolutely highlight the limits of their approach, because the transposition is not obvious. In many developing countries, solutions may have been identified without the possibility of implementing them, which is not necessarily the case in Australia.

Author Response

Thanks for your kind review and positive feedback.

Response:

The first point re sustainability: For this research, sustainability is embedded in all perspectives of the farming systems – use of land and water resources, economics, transportation of farm produce, and overall sustainability of the farming enterprises in northern Australia, particularly of small-scale farmers. Some changes included throughout the text, please see lines 598-600, 617-640, 709-718.

Regarding the second point, a major challenge for this research, included in lines 545-547, is to convey farmers’ perspectives to policy decision-makers. Implementation is an issue in Australia too.

Kind regards

Kamaljit

Reviewer 3 Report

The manuscript can be of interest to wide readers of journals and contributes to existing knowledge on the subject matter. However, I have pointed out few pertinent points for improving the clarity of the content and boosting the scientific soundness of the manuscript.

Abstract: Major findings of the study are missing in the abstract.

Figure 1,2,3 are missing

Line 409, 492: (Appendix 1-3), (Appendix 1-4). are missing

Author Response

Abstract: Major findings of the study are missing in the abstract.

Response:

  • Changes included (lines 32-35)

Figure 1,2,3 are missing

  • Files included

Line 409, 492: (Appendix 1-3), (Appendix 1-4). are missing

  • Appendices included.

Thanking you very much for supportive and positive feedback.

Kind regards

Kamaljit

Round 2

Reviewer 2 Report

Thank you for the updates.